# Do Bernstein’s Stages of Learning Apply after Stroke? A Scoping Review on the Development of Whole-Body Coordination after Cerebrovascular Accidents

**DOI:** 10.3390/brainsci13121713

**Published:** 2023-12-13

**Authors:** Anna Wargel, Steven van Andel, Peter Federolf

**Affiliations:** 1Department of Sport Science, University of Innsbruck, 6020 Innsbruck, Austria; anna.wargel@uibk.ac.at (A.W.);; 2IJsselheem Foundation, 8266 AB Kampen, The Netherlands

**Keywords:** motor control, stroke, coordination, degrees of freedom, injury management

## Abstract

Stroke is one of the leading causes of disability around the world, presenting unique challenges in motor development during the rehabilitation process. Based on studies in movement and sports science, thorough knowledge has accumulated on the development of movement skills. Through the works of Nikolai Bernstein, it has been established that when learning new skills, people tend to first simplify coordination by ‘freezing’ their degrees of freedom, after which they start building efficiency by ‘releasing’ specific degrees of freedom. If a similar pattern of development can be established post-stroke, it would imply that lessons learned in sports skill acquisition can also be implemented to optimize stroke rehabilitation. The current scoping review aims to assess whether the Bernsteinian freezing-to-releasing stages of learning also apply to developing whole-body movement skills after stroke. To this end, we systematically screened the existing literature for studies involving a longitudinal measure of whole-body coordination after a stroke. Only five articles met the criteria for inclusion, indicating a gap in research on this topic. Based on the observations within these articles, we could neither confirm nor reject whether the freezing-to-releasing process can apply after a stroke. We could, however, hypothesize a detailed description of the freezing-to-releasing process, which can be assessed in future works.

## 1. Introduction

Stroke is a major cause of death and disability, with a high impact on healthcare costs, particularly in low- and middle-income countries [1]. The duration of rehabilitation after stroke can vary depending on the nature and severity of the symptoms, as particularly severe motor symptoms may require re-learning gait and balance. Even after rehabilitation, movement problems may persist, leading to increased rates of falls [2], which may last for years after the event [3].

In our current work, we approach this persistence of movement problems as a skill acquisition problem. The field of movement control research has made impressive developments in recent decades, including a more thorough understanding of how the central nervous system works to control the body [4,5,6] and how we learn new movement skills [7,8]. These insights might have important implications for optimizing the rehabilitation process, particularly in the highly complex case of whole-body coordination.

The work of Nikolai Bernstein is often cited as a starting point for modern theories of motor control and learning [9]. He famously described the role of variability in movement as something other than an unwanted error: even experienced blacksmiths use a variety of hammer trajectories and yet consistently hit the anvil in the right place. Observations like these have led to a new perspective on movement variability. That is, variability might be allowed in the system as long as it does not interfere with the goal of the task. In fact, movement variability has been found to be functional as it generates knowledge about the current state of the movement system [10,11].

Another important fruit of Bernstein’s work was his description of how variability in movement or ‘degrees of freedom’ (DoF) change over the course of motor development. Bernstein described how novices struggling with the accuracy demands of a movement often ‘freeze’ DoF to simplify the coordinative problem. That is, early learners tend to fixate on certain joints (e.g., a young child throwing a ball fixates the torso and only uses the arm) or create a strong temporal coupling between movements across different joints in order to lower the skill’s complexity [12]. Once these freezing strategies are discovered, they enable the stable execution of movement, allowing the learner to gradually release DoF and enhance efficiency by involving more individual joints. This leads to the development of system degeneracy, i.e., the capacity to use the same system components or DoF to generate various movement solutions or coordinative patterns [13,14]. As a result, a flexible and adaptable movement pattern can be achieved.

Some studies of people without disabilities support this hypothesis with the use of a freezing/releasing strategy in motor learning [12,15,16,17], while others refute it [18,19,20]. It can be explained that the initial strategy of DoF freezing is neither exclusive nor ubiquitous and that the reorganization of DoF relies on the skill type, its objective, the individual’s constraints in performing the skill, and the environment in which it is executed [21]. Guimarᾶes and colleagues conducted a review study, demonstrating that research on the DoF freeze/unfreeze approach showed consistent results when analyzing skills in the same class with the same objective [22]. However, they also noted that there have been only a limited number of studies on this topic, and most had inconsistent designs.

Although there is not yet a unified scientific consensus on the process of freezing and releasing DoF, a comprehensive body of knowledge on this topic has been developed and documented. Furthermore, sports science coaching guidelines have started to take these findings to heart [7,23,24]. Practice guidelines have been developed which prescribe a manipulation of constraints [25], which “allows the individual to explore available system degeneracy by harnessing self-organization processes” [24] (p. 10), helping coaches and trainers develop flexible and adaptable movement patterns in their athletes.

The aim of the current scoping review is to assess whether the concepts of freezing-to-releasing DoF can also be applied to the rehabilitation process after a stroke. A clear relevance can be found in this study as follows: if the Bernstinian freezing-to-releasing DoF progression can be established when re-learning whole-body coordination after stroke, then it is plausible that the same learning guidelines can be applied to improve rehabilitation outcomes and ultimately increase mobility and decrease the risk of fall in stroke survivors. The following aims were formulated for the current review study: (1) to assess the scope of the research field investigating freezing-to-releasing DoF in stroke survivors. To achieve this aim, a focus on whole-body movement was chosen because, firstly, these movements seem most relevant for mobility and fall prevention, and second, in order to observe changes in coordination, many DoF need to be involved in the movement [26]. Furthermore, a ‘scoping review’ was deemed the most suitable design for achieving this aim. (2) As a second aim, we strive to assess whether we can find support for the premise that the re-development of whole-body skills after stroke follows a freezing-to-releasing DoF pattern.

## 2. Methods Section

### 2.1. Search Strategy

To better understand the scope of the research field, a systematic search was performed in PubMed and Web of Science. PubMed was chosen for its common usage in the medical field, which fitted well with the clinical focus of the review, while Web of Science was added for its wide range across disciplines and in human movement science in particular. The search was conducted in July 2022 and updated in September 2023. Our complete search strategy was designed to identify a broad spectrum of relevant research studies. The strategy consisted of combining keywords related to ‘stroke’, ‘(movement) variance’, and whole-body skills such as gait and posture, using the ‘AND’ operator. Expanding the scope of our review, we also integrated terms related to the freezing-to-releasing DoF concept, which encompassed variance, variability, uncontrolled manifold, and the minimal intervention principle (these latter two terms were added as they are part of approaches commonly used to assess movement variability). These supplementary terms were integrated into our search strategy to capture valuable insights into the variability and variance of movement, which can potentially serve as indicators of DoF freezing or releasing. For a detailed breakdown of our search strategy, please refer to Table 1. In undertaking this systematic review, we adhered to the Preferred Reporting Item for Systematic Reviews and Meta-Analyses Extension for Scoping Reviews (PRISMA-ScR) guidelines [27].

### 2.2. Eligibility Assessment

Only primary peer-reviewed sources were considered, which involved human subjects. Relating to the aim of describing the freezing-to-releasing DoF process, only studies that involved longitudinal measures across several days were included, and only if they involved some measure of whole-body coordination. To further specify when a movement constitutes ‘whole-body coordination’, the authors used the heuristic that if the movement included the torso as well as at least one limb, the study was included (e.g., re-learning a single arm movement was to be excluded, while re-learning an arm movement while torso posture was measured was included). Eligibility was assessed by two authors (AW and SA) independently in the phases of duplicate removal, title, and abstract screening. A full-text assessment of the papers was performed by one author (SA) and corroborated by the others. The update of the search and the eligibility assessment of the additional studies of all phases was carried out by one author (AW) and corroborated by the others. The management of the literature was carried out using Citavi Software (Version 6.17.0.0, Swiss Academic Software GmbH, Wädenswil, Switzerland).

### 2.3. Data Extraction and Synthesis

One author (SA) read all the full texts and extracted the methodological characteristics and main findings. Extracted data comprised the title, publication year, the name of the first author, study design, main aim, sample size, and population description. Additionally, we collected information on the assigned task, the primary outcomes, variability outcomes, and whether a description of DoF was provided. For this purpose, a Google spreadsheet form was devised and utilized to compile the data from the included studies. For the update, another author (AW) took over this process. The other two authors in each case carefully scrutinized this process to ensure the validity of the findings. As it was considered unlikely that many studies would literally report on a Bernsteinian freezing-to-releasing process, the authors made their own inferences and interpretations as to the meaning of the study results for this process.

### 2.4. Assessment of Methodological Quality

To assess the methodological quality of the studies, we utilized a modified version of the checklist for randomized and non-randomized studies by Down and Black [28]. The quality assessment was conducted by one author (AW) and monitored by another author (PF). The modified version contained 15 items. Items referring to targeted interventions and associated confounding factors were excluded since the studies included in the present study are longitudinal studies without explicit intervention. Questions receiving affirmative responses were assigned a score of 1, while all others were assigned a score of 0. The maximum possible score was 15. The study quality increases as the score increases.

## 3. Results

### 3.1. Article Selection

In total, 1086 hits were found in the two search engines at two different timepoints (initial search and pre-submission updated search). After screening steps, five articles met the criteria for inclusion in the review (Figure 1). One article was excluded after abstract screening because the full text could not be obtained online or by contacting the authors [29].

### 3.2. Methodological Characteristics

The five included studies were all of different designs and included stroke survivors of differing times after their stroke (ranging from less than 1 to more than 6 months post-stroke). Some consistency was established in the included studies, as three studies used an optical motion capture system, and two of these combined this with an uncontrolled manifold (UCM) analysis. For reference, Table 2 presents the methodological specifications of the five included studies, including patient characteristics, the timing of inclusion, tasks, timepoints, methods, and dependent variables.

### 3.3. Results Related to Variability in Movement Trajectory

A presentation of the key findings, the outcomes associated with movement variability, and the interpretation of results concerning the freezing and releasing of DoF are shown in Table 3. Three studies made mention of (changes in) movement variability during rehabilitation after stroke, two of which were identified using the UCM analysis and one by describing the number of joints (degrees of freedom) involved in completing the required action.

### 3.4. Quality Assessment

All studies included in this scoping review used a reasonable methodology. The aggregate scores ranged from 8 to 13 out of a total of 15 points. Notable common limitations included a lack of details on the characteristics of the population from which subjects were recruited, the participant selection method, or the number of individuals who consented to participate. The location where measurements were conducted and their representativeness were rarely documented. None of the studies performed a power analysis.

## 4. Discussion

The current study had the following two aims: firstly, to evaluate the extent of research on whole-body coordination development after stroke, and secondly, to assess the applicability of the Bernsteinian freezing-to-releasing DoF concept in this context. The search identified only five eligible studies that examined the longitudinal assessment of whole-body movement coordination development, indicating a gap in the knowledge. This aligns with prior review studies of motor-learning capabilities after stroke, which either focused only on the upper limb [34] or found that very few learning studies included whole-body skills [37]. Among these five studies, three reported measures of movement variability that could be related to the Bernsteinian concept of freezing and releasing DoF. From the interpretation of the authors, this led to four observations regarding changes in DoF, two of which were consistent with the concept of the freezing-to-releasing DoF concept, and two were found to be inconsistent.

In the interpretation of the authors, two observations in the original studies were found to corroborate the freezing-to-releasing DoF progression, and two were found to be inconsistent with this concept. Discussing these inconsistencies, in the study of Papi and colleagues [32], which is based on the uncontrolled manifold approach (UCM), the disappearance of a peak in variability in the UCM (V_UCM_) at a late stance when walking without ankle–foot orthosis (AFO) between 3 and 6 months after the baseline is inconsistent with Bernsteinian’s theory of releasing DoF. In releasing DoF, one can expect V_UCM_ to grow, as this is indicative of a greater coordinative repertoire that is able to perform the same task (indicative of system degeneracy). The original study provides few details that explain this finding as the interval between measures is large (3 months), and no report is made as to the activity of the single participant between sessions. For example, if the participant always walks with his AFO between sessions, then after 6 months, walking without AFO presents as a less-practiced task, which might elicit the new freezing of the DoF response. The second inconsistency with the Bernsteinian freezing-to-releasing concept was found in the study of Roby-Brami and colleagues [34], where more affected patients seemed to engage their torso in making reaching movements rather than ‘freezing’ these additional DoF. However, it is possible that spasticity at the elbow (as indicated in Figure 3 in [34]) may have contributed to this outcome, where the limited range of motion can only be compensated for by recruiting additional DoF. Since spasticity measures or scaling to personal ability were not part of the study, more research is required to further explore these findings. Together, these findings emphasize the influence of individual constraints (e.g., personal experience/spasticity) on the emerging coordination. A stroke can result in significant alterations to these limitations, prompting the individual to adopt compensatory movement strategies. Over time, these limitations to individual constraints may gradually improve while the learning stages simultaneously evolve. This can complicate independent research of these processes. As a result, forthcoming studies should outline these individual constraints to evaluate their impact on the learning process.

The strongest observation in support of the freezing-to-releasing progression comes from a study involving 10 people post-stroke [31]. They showed an increase in V_UCM_ at late swing, which is indicative of improvements in system degeneracy. The study included two groups of participants (<1 month after stroke vs. >6 months), but no group effects were reported, leaving uncertainty as to when this ‘releasing of DoF’ occurs during the rehabilitation process. It is interesting that Lofrumento and colleagues reported a consistently low UCM ratio, with variability orthogonal to the UCM (V_ORT_, ‘bad variability’) being higher than V_UCM_ (‘good’ or ‘functional variability’). We reason here that this could also be interpreted as consistent with a freezing strategy, albeit one that has been affected by the stroke. In the UCM approach, V_ORT_ is thought of as ‘bad variability’ because it can interfere with the task’s goal, while V_UCM_ can exist without affecting the task’s outcome. A healthy movement system learns how to control and exploit V_UCM_ and minimizes V_ORT_ to create an adaptable and error-free movement pattern. The result of Lofrumento et al. could be explained as a freezing strategy in which the person still needs to (re-)learn how to minimize V_ORT_ and, therefore, only controls and minimizes V_UCM_. This pattern has been previously described in patients with multiple sclerosis [37], and similar patterns have been observed in other neurological disorders [38].

Some limitations need to be considered when interpreting the findings of the current study. As is usual in the scoping literature review, we designed an a priori search and selection strategy. Elements of this strategy, such as the inclusion of only two search engines and the strict definition of what we consider to be ‘whole-body movement’ or ‘longitudinal measures’, might have limited the number of included studies. It needs to be acknowledged that types of different decision making in the early stages might have resulted in a higher number of included studies, and some level of subjectivity is inherent to the design of this type of study. However, even with different search terms, it is unlikely that the amount of included studies would increase so much that our conclusion would change; more research is required to study whole-body motor development after stroke from the perspective of Bernstein’s learning stages.

## 5. Conclusions

With only five studies meeting the inclusion criteria for this review, resulting in only four observations that could be related to the Bernsteinian process of freezing-to-releasing DoF, the main outcome of this study is that little is known about the development of whole-body coordination after stroke. Further research is recommended on the general topic of whole-body coordination development. Based on the observations of this review, we can neither confirm nor reject the freezing and releasing of DoF stages as described by Bernstein. Future studies could investigate the freezing-to-releasing DoF process as follows: (1) an overall (good and bad) variability increase resulting from the stroke, (2) a freezing strategy minimizes either or both good and bad variability and (3) releasing DoF is characterized by a gradual increase in good variability, increasing system degeneracy. The timing of progression between phases may vary significantly and relies on stroke severity and personal characteristics. For this reason, future studies should describe changes in movement variability at regular intervals, taking into account individual differences in progression and development.

## Figures and Tables

**Figure 1 brainsci-13-01713-f001:**
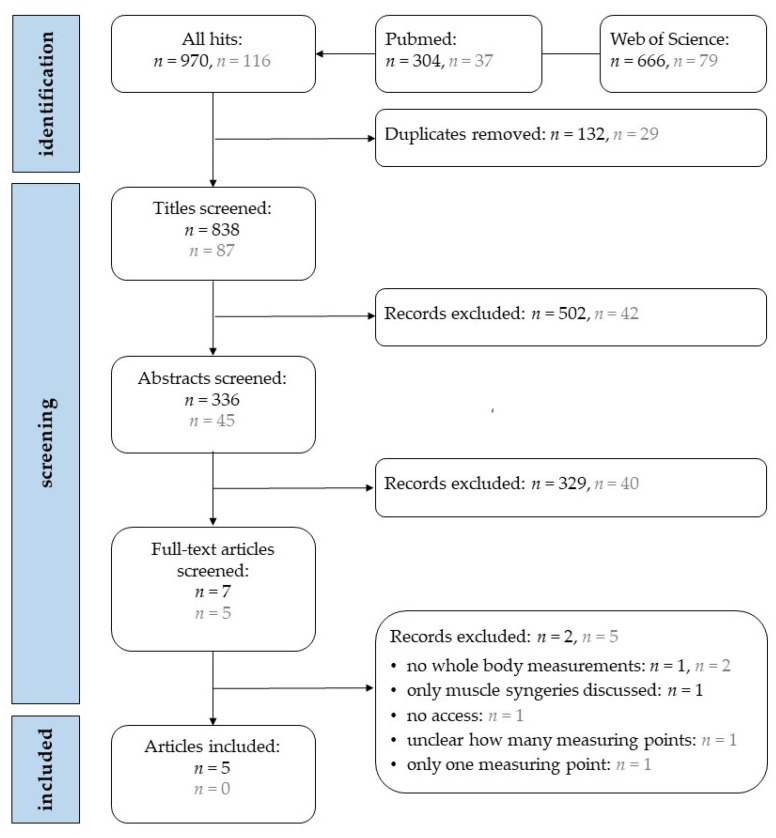
Flowchart of article selection from the first search (black) in July 2022 and the second search (grey) in September 2023.

**Table 1 brainsci-13-01713-t001:** Details of systematic search.

Search Component	PubMed	Web of Science
Stroke	((stroke rehabilitation [MeSH Major Topic]) OR (stroke [Title/Abstract])) OR (“Cerebrovascular accident” [Title/Abstract])	Stroke OR “Cerebrovascular accident”
	AND	AND
(Movement) variance	“Variance” [Title/Abstract] OR “Uncontrolled Manifold” [Title/Abstract] OR “Minimal Intervention” [Title/Abstract] OR “freezing” [Title/Abstract] OR “synerg*” [Title/Abstract] OR “Degrees of Freedom” [Title/Abstract] OR Bernstein* [Title/Abstract]	Variance OR “Uncontrolled Manifold” OR “Minimal Intervention” OR freezing OR synerg* OR “Degrees of Freedom” OR Bernstein*
	AND	AND
Whole-body skills	“whole-body” [Title/Abstract] OR “whole-body” [Title/Abstract] OR “gait” [Title/Abstract] OR “posture” [Title/Abstract] OR “locomotion” [Title/Abstract]	“whole body” OR “whole-body” OR gait OR posture OR locomotion

* serves as a place holder for zero or more characters except blank space (e.g. synerg*–synergies)

**Table 2 brainsci-13-01713-t002:** Methodological characteristics of the included studies.

	Participants and Time of Inclusion	Task	Timepoints	Methods	Dependent Variables
Caty et al. (2009) [30]	*n* = 10, >6-month post-stroke	2 min treadmill walk	3 measures: test repeated 1 day and 1 month after baseline	Optical motion capture and oxygen consumption measures	Lower limb kinematic variables
Lofrumento et al. (2021) [31]	*n* = 6, <1-month post-stroke;	3 min treadmill walk and 6 steps overground	Tests before and after 4 weeks of conventional therapy	Optical motion capture with UCM analysis method ^a^	Variance of UCM and the orthogonal subspace of the ankle joint trajectories
	*n* = 4, >6-month post-stroke				
Papi et al. (2015) [32]	*n* = 1, 2 months post-stroke;	6 repeats of 6 meters of overground walking	3 measures: test repeated 3 and 6 months after baseline	Optical motion capture with UCM analysis method ^a^	Variance of UCM and the orthogonal subspace of lower limb sagittal joint kinematics
	*n* = 6, healthy controls				
Shin et al. (2020) [33]	*n* = 9, <1-month post-stroke;	Measures during conventional therapy	5 to 12 sessions over the course of rehabilitation	Inertial motion capture to extract amount of activity information	Amount of motion (total amount of joint displacements measured from inertial motion capture)
	Subgroup of *n* = 6 was longitudinally monitored				
Roby-Brami et al. (2003) [34]	*n* = 6, single measure, 48–162 days post-stroke;	Seated reaching movements	1 to 3 measures at monthly timepoints during regular therapy	Electromagnetic motion tracking	Peak velocity of the hand and movement duration, amount of acromion displacement, and joint angular variations
	*n* = 9, repeated measures, 24–89 days post stroke;				
	*n* = 7 healthy controls				

^a^ UCM: uncontrolled manifold [35,36].

**Table 3 brainsci-13-01713-t003:** Results relating to variability movement trajectories.

	Main Study Outcomes	Outcomes Related to Movement Variability and the Freezing-to-Releasing Process	Interpretation of Results Relating to Freezing and/or Releasing DoF
Caty et al. (2009) [30]	Good reliability was established for the kinematic gait variables among stroke patients.	NR	
Lofrumento et al. (2021) [31]	UCM hypothesis rejected: V_ORT_ consistently greater than UCM variability.	No treatment/time effects, except a slight increase in V_UCM_ at the end of the swing phase.	The increase in V_UCM_ at late swing is **consistent** with releasing DoF.
Papi et al. (2015) [32]	UCM method successfully applied: with rehabilitation, the UCM ratio becomes ‘like normal’ when using an ankle–foot orthosis. More research is required to confirm the findings from this N = 1 study.	Stroke patients consistently show higher V_UCM_ than V_ORT_ with progression towards a ratio waveform similar to healthy controls at 6 months.	Both V_UCM_ and V_ORT_ strongly decrease between the baseline and 3-month follow-up, **consistent** with a freezing DoF. In measures without an ankle–foot orthosis, a peak in V_UCM_ disappears between 3 and 6 months, which is **inconsistent** with releasing DoF.
Shin et al. (2020) [33]	A relationship was established between the amount of movement around the lower limb joints and gait speed at follow-up.	NR	
Roby-Brami et al. (2003) [34]	Stroke patients established different levels of coordination in reaching compared to healthy controls.	Patients with greater impairment recruited additional DoF (trunk bending) to compensate for the limited range of motion in distal joints.	Recruitment of more trunk activity is **inconsistent** with freezing DoF.

NR: None reported; V_UCM_: variability in the uncontrolled manifold, which does not impact task performance; V_ORT_: variability orthogonal to the uncontrolled manifold, which does affect the task performance; marked in bold: highlights the main result of this study.

## Data Availability

Not applicable.

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
