# Peer review of "Do Bernstein’s Stages of Learning Apply after Stroke? A Scoping Review on the Development of Whole-Body Coordination after Cerebrovascular Accidents"

_brainsci, 2023, doi:10.3390/brainsci13121713_

Round 1
Reviewer 1 Report
Comments and Suggestions for Authors
Overall, this review paper is well-written in terms of flow and fluency. However, my major criticism is that the scoping review has not followed the PRISMA guidelines. I would like the authors to revise the article as per the PRISMA-ScR checklist, specifically designed for scoping reviews (https://www.equator-network.org/reporting-guidelines/prisma-scr/). Doing so, I believe the quality will further improve.
Right now, some of the concerns are with the PRISMA flowchart. There is no breakdown of the reasons why a majority of the articles were excluded. This information is vital to judging the quality of the scoping review. I urge the authors to go through the guidelines and provide a rebuttal on how every item of the PRISMA-ScR checklist has been addressed in the revision.
Notwithstanding the PRISMA guidelines, I think the authors will still need to provide details of the stroke phase (acute/sub-acute/chronic) vs. type (ischemic/hemorrhagic) as they can affect the overall conclusion.
I am sure the authors have done their due diligence, and the writing quality is pretty good. Although the conclusion that the Bernsteinian theory can neither be accepted nor be refuted is a bit disappointing, this review article can potentially help future stroke research to consider whole-body coordination, especially in gait-related studies.
Author Response
Dear Reviewer,
thank you for your time and for critically assessing the manuscript! We revised the paper and incorporated your comments as described below. All changes are highlighted in red in the manuscript.
In the name of all authors,
Sincerely,
Peter Federolf
Comment:
Overall, this review paper is well-written in terms of flow and fluency. However, my major criticism is that the scoping review has not followed the PRISMA guidelines. I would like the authors to revise the article as per the PRISMA-ScR checklist, specifically designed for scoping reviews (https://www.equator-network.org/reporting-guidelines/prisma-scr/). Doing so, I believe the quality will further improve.
Right now, some of the concerns are with the PRISMA flowchart. There is no breakdown of the reasons why a majority of the articles were excluded. This information is vital to judging the quality of the scoping review. I urge the authors to go through the guidelines and provide a rebuttal on how every item of the PRISMA-ScR checklist has been addressed in the revision.
Response:
Many thanks for your valuable feedback. Please find the completed PRISMA-ScR checklist attached. We would like to comment on a few items:
- Item 3:
added “Furthermore, a 'scoping review' was deemed the most suitable design for achieving this aim. (2)” in the introduction (see line 86 f.) - Item 5: We did not perform a registry, although we agree with you that this is important, especially for systematic reviews. Whilst we did create a study protocol in preparation of the study, we have not published this as we have not seen this as a requirement for scoping reviews. Unfortunately, we can no longer change this point, as this should have been done in advance.
- Item 10: We added the form used (see line 135 f.)
- Item 11: We added all extracted data items (see line 132 ff.)
- Item 12: We have not carried out a methodological quality assessment, as this work is a scoping review and is therefore intended to provide an overview. In principle, it is of course important to check the methodological quality. However, it would not help us to achieve our aims in this case. In fact, the PRISMA guide (https://www.acpjournals.org/doi/full/10.7326/M18-0850) states: "A key difference between scoping reviews and systematic reviews is that the former are generally conducted to provide an overview of the existing evidence regardless of methodological quality or risk of bias (4, 5). Therefore, the included sources of evidence are typically not critically appraised for scoping reviews."
- Item 14: We added the reasons for the excluded articles in the PRISMA flowchart.
- Item 16: see response to Item 12
Comment:
Notwithstanding the PRISMA guidelines, I think the authors will still need to provide details of the stroke phase (acute/sub-acute/chronic) vs. type (ischemic/hemorrhagic) as they can affect the overall conclusion.
Response:
We did not limit ourselves to specific categorizations in this review as to not limit the search results. Furthermore, in these results we opted to stick to information about the sample that was available in all studies (time since stroke) to be able to report on a consistent measure. If we would have added more stroke details this would have led to incompleteness in our reporting of the results, which we believe, would decrease readability.

Reviewer 2 Report
Comments and Suggestions for Authors
This is an exciting research paper.
However, a few suggestions are placed to further improve the manuscript.
Introduction:
Comment 1: Nicely written. However, aim should be very precise. The length of the introduction too big to comprehend, kindly cut it short.
Method:
Results:
Comment 2: looks good
Discussion:
Comment 3: The first paragraph may be modified. It is always better to start a discussion with the important findings of your result and the context in which it will be important or valuable than the already existing data. Rest of the discussion part is well described.
Reference:
Comment 5: Looks good
Table and Figure:
Comment 6: looks good
Author Response
Dear Reviewer,
thank you for your time and for critically assessing the manuscript! We revised the paper and incorporated your comments as described below. All changes are highlighted in red in the manuscript.
In the name of all authors,
Sincerely,
Peter Federolf
Introduction: Comment 1:
Nicely written. However, aim should be very precise. The length of the introduction too big to comprehend, kindly cut it short.
Response:
Thank you for the kind assessment of the introduction. Since the editor has commented that the current manuscript is already on the short side for this publication type, with the suggestion of elongating it, we have opted not to shorten the introduction further. However, we feel that this comment might also indicate the feeling that the narrative might sometimes wander away from the direct aim of the study. As such, we have made edits throughout to strengthen the foundation behind the aims of our study.
Discussion: Comment 3:
The first paragraph may be modified. It is always better to start a discussion with the important findings of your result and the context in which it will be important or valuable than the already existing data. Rest of the discussion part is well described.
Response:
Thank you very much for your comment, we have followed your suggestion and changed it.
Reviewer 3 Report
Comments and Suggestions for Authors
Review brainsci-2699104-peer-review-v1
In the paper titled 'Do Bernstein’s stages of learning apply after stroke? A scoping review on the development of whole-body coordination after a cerebrovascular accident,' two interesting research objectives are presented. The first objective aims to assess the scope of research in the field investigating the freezing-to-releasing Degrees of Freedom (DoF) in stroke survivors. The second objective aims to assess whether there is support for the premise that the re-development of whole-body skills after a stroke follows a freezing-to-releasing DoF pattern. While the paper is well-written, I have some comments and suggestions, listed below in the order they appear in the text rather than their importance.
I'm wondering if the paper's title could be shorter. Here's a suggestion: Do Bernstein's Learning Stages Apply After a Stroke (Cerebrovascular Accident)? - A Scoping Review.
Line 4 – Please remove the period at the end of the title.
Line 5 – For Steven van Andel, it appears that the affiliation should be number 12. It's a good idea to separate the numbers with commas.
Line 108 – Table 1 should be moved to this line to enhance the paper's readability.
Why did the authors limit themselves to only two databases and not include Science Direct and EBSCO?
The paper is missing the Quality Assessment chapter. It might be beneficial to use the study: Downs, S. H., and N. Black. "The Feasibility of Creating a Checklist for the Assessment of the 538 Methodological Quality Both of Randomised and Non-Randomised Studies of Health Care 539 Interventions." J Epidemiol Community Health 52, no. 6 (1998): 377-84.
The Results section should be organized into subsections for better structure. The first subsection, titled i.e. 'Methodological Characteristics of the Included Studies,' should be placed immediately after Figure 1. This section should provide a more detailed discussion of the information found in lines 136 - 138. What is meant by 'Time points'? Are there repeated tasks or study methods or 'Time of inclusion'? Additionally, the last column of the reference table should be linked to the first column.
The next subsection should be entitled i.e. 'Results Related to Variability in Movement Trajectories.' Here, I have the same comments as mentioned above; the Table 3 description is missing.
Author Response
Dear Reviewer,
thank you for your time and for critically assessing the manuscript! We revised the paper and incorporated your comments as described below. All changes are highlighted in red in the manuscript.
In the name of all authors,
Sincerely,
Peter Federolf
Comment:
I'm wondering if the paper's title could be shorter. Here's a suggestion: Do Bernstein's Learning Stages Apply After a Stroke (Cerebrovascular Accident)? - A Scoping Review.
Response:
We thank the reviewer for this comment and we see that shorter titles often bring more clarity. However, in this case, we found that shortening the title leads to the omission of some vital characteristics of the study. We consider Bernstein, cerebrovascular accident (stroke) and whole-body coordination equally important in the design of this study and we were unable to come up with a shorter title that brings justice to all three elements.
Comment:
Line 4 – Please remove the period at the end of the title.
Response: Done
Comment:
Line 5 – For Steven van Andel, it appears that the affiliation should be number 12. It's a good idea to separate the numbers with commas.
Response: Done
Comment:
Line 108 – Table 1 should be moved to this line to enhance the paper's readability.
Response: Agreed
Comment:
Why did the authors limit themselves to only two databases and not include Science Direct and EBSCO?
Response:
We agree with the reviewer that for a comprehensive literature review, one needs to include more than two databases. However, considering the scoping nature of this work, we deemed two databases to be sufficient in achieving these aims. Especially because we deemed these two databases to be the most promising in the two research fields we aim to combine (Pubmed for the medical field and Web of Science for movement science).
Comment:
The paper is missing the Quality Assessment chapter. It might be beneficial to use the study: Downs, S. H., and N. Black. "The Feasibility of Creating a Checklist for the Assessment of the 538 Methodological Quality Both of Randomised and Non-Randomised Studies of Health Care 539 Interventions." J Epidemiol Community Health 52, no. 6 (1998): 377-84.
Response:
You are right, especially with regard to systematic reviews. However, as this is a scoping review with the aim of providing an overview, we have decided not to carry out a quality assessment. It would not help us to achieve our aims in this case. In fact, the PRISMA guide (https://www.acpjournals.org/doi/full/10.7326/M18-0850) states: "A key difference between scoping reviews and systematic reviews is that the former are generally conducted to provide an overview of the existing evidence regardless of methodological quality or risk of bias (4, 5). Therefore, the included sources of evidence are typically not critically appraised for scoping reviews."
Comment:
The Results section should be organized into subsections for better structure. The first subsection, titled i.e. 'Methodological Characteristics of the Included Studies,' should be placed immediately after Figure 1. This section should provide a more detailed discussion of the information found in lines 136 - 138. What is meant by 'Time points'? Are there repeated tasks or study methods or 'Time of inclusion'? Additionally, the last column of the reference table should be linked to the first column.The next subsection should be entitled i.e. 'Results Related to Variability in Movement Trajectories.' Here, I have the same comments as mentioned above; the Table 3 description is missing.
Response:
Thank you for this good comment. We agree and have adjusted the results section.
Reviewer 4 Report
Comments and Suggestions for Authors
Thanks for the opportunity to review this interesting manuscript. After to revise it, I ask the authors for the following concerns:
The first sentence of the introduction is not necessary. Later, authors can provide epidemiological data and cite the most important approaches in rehabilitation, for example physiotherapy.
What are the main causes of limitations in movement control?
Please, replace "neural system" by "central nervous system".
I think tha literature search must be performed in more databases, such as Scopus, WOS or CINAHL Complete.
Line 110-111: Restriction in language (only studies writeen in English, German or Dutch). This is a problem in your search. Probably you have lost a lot of studies in other languages. This is a red flag for your review.
What scale have you used to assess the methodological of the studies included?
Table 1 must appear in methods.
I consider that the description of the results or findings of your scoping review is very vague, so inexistent. Please, write more about this. This is the most important.
You must write a conclusion in a separate paragraph.
Author Response
Dear Reviewer,
thank you for your time and for critically assessing the manuscript! We revised the paper and incorporated your comments as described below. All changes are highlighted in red in the manuscript.
In the name of all authors,
Sincerely,
Peter Federolf
Comment:
The first sentence of the introduction is not necessary. Later, authors can provide epidemiological data and cite the most important approaches in rehabilitation, for example physiotherapy.
Response: Agreed. We deleted the first sentence.
Comment:
What are the main causes of limitations in movement control?
Response:
We now consistently refer to this as 'movement problems' in both the first and second paragraphs - to make it clear that we are simply building on the previous paragraph. We have also changed 'motor' to 'movement' because the use of motor creates a redundant distinction between sensory and motor systems. Movement' is a more holistic term, encompassing both perceptual and motor processes.
Thank you for pointing this out.
Comment:
Please, replace "neural system" by "central nervous system".
Response: Agreed
Comment:
I think the literature search must be performed in more databases, such as Scopus, WOS or CINAHL Complete.
Response:
We agree with the reviewer that for a comprehensive literature review, one needs to include more than two databases. However, considering the scoping nature of this work, we deemed two databases to be sufficient in achieving these aims. Especially because we deemed these two databases to be the most promising in the two research fields we aim to combine (Pubmed for the medical field and Web of Science for movement science).
Comment:
Line 110-111: Restriction in language (only studies writeen in English, German or Dutch). This is a problem in your search. Probably you have lost a lot of studies in other languages. This is a red flag for your review.
Response:
We appreciate that in theory some articles might have been missed due to these language constraints. However, as many published literature reviews only include English articles, and the international scientific language is English, we see the inclusion of English language articles as the basis of the review and the incorporation of German and Dutch studies as added benefit. We do not consider ourselves master enough of any other languages to allow us to read, critically reflect on, and interpret articles in any other languages.
Comment:
What scale have you used to assess the methodological of the studies included?
Response:
We have not carried out a methodological quality assessment, as this work is a scoping review and is therefore intended to provide an overview. In principle, it is of course important to check the methodological quality. However, it would not help us to achieve our aims in this case. In fact, the PRISMA guide (https://www.acpjournals.org/doi/full/10.7326/M18-0850) states: "A key difference between scoping reviews and systematic reviews is that the former are generally conducted to provide an overview of the existing evidence regardless of methodological quality or risk of bias (4, 5). Therefore, the included sources of evidence are typically not critically appraised for scoping reviews."
Comment:
Table 1 must appear in methods.
Response: Done
Comment:
I consider that the description of the results or findings of your scoping review is very vague, so inexistent. Please, write more about this. This is the most important.
Response: We have revised the results section and hope this is in line with your request.
Comment:
You must write a conclusion in a separate paragraph.
Response:
We followed your recommendation and added a conclusion section.
Round 2
Reviewer 3 Report
Comments and Suggestions for Authors
The scoping review titled "Do Bernstein’s stages of learning apply after stroke? A scoping review on the development of whole-body coordination after cerebrovascular accident" is a compelling and interesting paper. I have no specific comments regarding the paper itself. However, I recommend that the authors evaluate the papers included in the review. I suggest utilizing the manuscript written by Downs and Black: The feasibility of creating a checklist for the assessment of the methodological quality both of randomised and non-randomised studies of health care interventions for this purpose.
Author Response
Dear reviewer, dear editor,
Thank you for your valuable comments. We have made an effort to reflect your suggestions in the manuscript. All changes are now highlighted in blue color (in addition to the red-color changes from R1). Please find below a point-by-point response to the comments.
Sincerely,
on behalf of all co-authors,
Peter Federolf
Reviewer Comment
The scoping review titled "Do Bernstein’s stages of learning apply after stroke? A scoping review on the development of whole-body coordination after cerebrovascular accident" is a compelling and interesting paper. I have no specific comments regarding the paper itself. However, I recommend that the authors evaluate the papers included in the review. I suggest utilizing the manuscript written by Downs and Black: The feasibility of creating a checklist for the assessment of the methodological quality both of randomised and non-randomised studies of health care interventions for this purpose.
Response
We greatly appreciate your positive feedback on our work.
In response to your recommendation regarding the quality evaluation of the included papers, we have decided to incorporate your suggestion. We modified the questionnaire developed by Downs and Black for assessing the quality of included studies to a 15-item appraisal sheet. The methodology and outcomes of this quality evaluation have now been integrated into our manuscript, with detailed commentary provided in lines 142-151 and lines 181-187.
We’ll attach the table with the specific paper-by-paper scores to this response.

Reviewer 4 Report
Comments and Suggestions for Authors
Thank you.
I consider that it is necessary to increase the number of databases to search and that restrictions in language are an important limitation.
Author Response
Dear reviewer, dear editor,
Thank you for your valuable comments. We have made an effort to reflect your suggestions in the manuscript. All changes are now highlighted in blue color (in addition to the red-color changes from R1). Please find below a point-by-point response to the comments.
Sincerely,
on behalf of all co-authors,
Peter Federolf
Reviewer Comment
I consider that it is necessary to increase the number of databases to search and that restrictions in language are an important limitation.
Answer:
Regarding the first issue, databases:
In R1 three databases were suggested: “Scopus, WOS or CINAHL Complete”. – We have used WOS in our search for papers, as suggested by the reviewer. In addition, we used PubMed to double-check that we did not miss a paper from the medical field.
Regrettably, due to institutional access limitations, we could not utilize Scopus or CINAHL Complete, as both are cost-prohibitive databases and are not available through our institutions. Furthermore, our current budget constraints preclude the possibility of purchasing access to additional databases.
We acknowledge that this limitation may hinder our ability to conduct a more exhaustive search, potentially missing some non-impact factor papers that might be listed in the aforementioned databases. Despite this, we maintain that our search methodology and the resultant findings adhere to the rigorous standards of scientific investigation and quality expected in a review within our field of study.
Regarding the second issue, language:
We have revisited our search results and identified all papers that were not written in English. Specifically, we found 6 such articles in PubMed and 9 in Web of Science. Upon re-evaluating these articles through title and abstract screenings, we confirmed that none of them met our other eligibility criteria. As per the reviewer's suggestion, we have now eliminated the language restriction as a criterion for paper inclusion in our review, since now all hits obtained through our search strategy were considered, irrespective of language. We have made the necessary adjustments to our manuscript (line 114).